# Quantifying Document Impact in RAG-LLMs

## Abstract

Retrieval Augmented Generation (RAG) enhances Large Language Models (LLMs) by connecting them to external knowledge, improving accuracy and reducing outdated information. However, this introduces challenges such as factual inconsistencies, source conflicts, bias propagation, and security vulnerabilities, which undermine the trustworthiness of RAG systems. A key gap in current RAG evaluation is the lack of a metric to quantify the contribution of individual retrieved documents to the final output. To address this, we introduce the Influence Score (IS), a novel metric based on Partial Information Decomposition that measures the impact of each retrieved document on the generated response. We validate IS through two experiments. First, a poison attack simulation across three datasets demonstrates that IS correctly identifies the malicious document as the most influential in 86% of cases. Second, an ablation study shows that a response generated using only the top-ranked documents by IS is consistently judged more similar to the original response than one generated from the remaining documents. These results confirm the efficacy of IS in isolating and quantifying document influence, offering a valuable tool for improving the transparency and reliability of RAG systems.

## 1 Introduction

Retrieval Augmented Generation (RAG) (13) is a natural language processing technique that enables large language models (LLMs) utilize external knowledge. By connecting LLMs to external, up-to-date knowledge sources, RAG systems can generate responses that are more accurate, current, and contextually relevant. This approach has demonstrated considerable promise in mitigating persistent LLM challenges such as knowledge obsolescence and improving factual grounding, as the models can reference verifiable information without necessitating costly retraining cycles. The appeal of RAG is widespread, largely due to its capacity to incorporate the latest information and ground outputs in attributable sources, thereby enhancing the utility of LLMs across diverse applications.

While RAG architectures aim to enhance reliability, they concurrently give rise to critical "explainability gaps", where the precise rationale behind a generated output remains opaque. LLMs within RAG frameworks can still produce "hallucinations" or "confabulations"—outputs that are inconsistent with retrieved facts or deviate from common sense, even when grounded (1; 17). Furthermore, issues such as "source confliction"—sources that have conflicting information, causing vague or contradictory responses (22; 16), the potential for "bias propagation" from skewed or unrepresentative retrieved documents (21; 7; 23), and "security vulnerabilities" including the injection or amplification of malicious content (19; 6; 27; 24; 29), continue to undermine system integrity. These problems collectively erode the trustworthiness and robustness of RAG systems, posing considerable risks, particularly in high-stakes domains such as healthcare, finance, and law. To better understand the way retrieval and generation interact, we need tools to analyze this interaction to make RAG more understandable.

A significant lacuna in current RAG research and evaluation methodologies is the absence of a metric to quantify how each individual retrieved document contributes to the LLM's final generated output. While numerous RAG evaluation frameworks assess crucial aspects like retrieval accuracy, contextual relevance, output faithfulness, and coherence, they often do not isolate the specific generative impact of single documents within the retrieved set. We introduce **Influence Score** (IS), a novel quantitative measure designed to

reflect the extent to which each specific retrieved document affects the final generated response. This score is derived using Partial Information Decomposition based on the excluded information metric (8).

To validate our proposed metric, IS, we conduct two sets of experiments. First, we perform a poison attack simulation using HotPotQA (25), Natural Questions (11), and MsMarco (18) as the datasets, and as PoisonedRAG (28) our poisoning method. After poisoning the retrieved documents to elicit incorrect answers, we use IS to measure each document's influence on the generation. In 86% of test cases, IS successfully identifies the poisoned document as the most influential.

Second, we perform an ablation study to evaluate the ability of IS to rank documents by importance. Using the same datasets, we began by generating a baseline response (**Response A**) using a full set of retrieved documents. We then ranked these documents with IS and created two new responses: **Response B**, generated from only the top two documents with the highest IS, and **Response C**, from the remaining documents. A panel of judges, consisting of both human evaluators and an LLM, were asked to determine whether **Response B** or **Response C** more closely matched the baseline. A preference for **Response B** would indicate that IS effectively identifies the most influential documents. In an overwhelming majority of cases, the judges selected **Response B**, confirming our metric's efficacy.

Furthermore, by having a framework that quantifies the impact of each retrieved document in the LLM response, we can pinpoint the documents responsible for each response. Specifically, it helps us with

- **Enhanced Source Attribution and Fact-Checking:** Giving each document a clear weight helps users track where information comes from and judge how trustworthy it is.

- **Model Calibration, and Identifying Bias:** Analyzing document impact will help us find out what content our LLM focuses on, and as result reveal potential biases in the knowledge base and the need for calibration.

- **Document Relevance Ranking:** By quantifying document impact, we can refine retrieval algorithms, improving the quality of retrieved documents and the overall response quality.

- **Adversarial Attacks and Model Poisoning:** If our LLM produces an undesirable response, we can easily locate the responsible document and remove the poisoned data.

## 2 Background and Motivation

### 2.1 RAG

The RAG process involves three components: an LLM, an external database, and a user query. The mechanism is as follows: First, we retrieve relevant documents from the database based on the user query (Retrieval). Calculating document relevance is an active area of research with various methods, such as cosine similarity between the document and query, Semantic Entropy (10; 15), and using another LLM for quantification (26). Next, we augment the user query with the retrieved information to create a more comprehensive prompt (Augmentation). Finally, we feed the augmented query to our LLM to generate a result (Generation). This approach enables the LLM to produce more informed, accurate, and contextually relevant responses by leveraging external knowledge in addition to its trained knowledge.

### 2.2 Safety of RAG LLMs

RAG LLMs are often considered a solution to the problem of "hallucination" in standard LLMs. By grounding their responses in retrieved documents, RAG models can provide more accurate and up-to-date information. However, this reliance on external data also introduces new and significant safety concerns that are not present in their non-RAG counterparts. While RAG models can be less prone to generating entirely fabricated information, they are more susceptible to a range of attacks that exploit their retrieval mechanism, making them, in some ways, less safe than standard LLMs (1).

One of the most significant vulnerabilities of RAG models is retrieval poisoning. In this type of attack, malicious actors contaminate the external knowledge base that the RAG model uses for retrieval. This can be done by injecting false, biased, or harmful information into the data sources. When the RAG model retrieves this poisoned data, it can be used to generate responses that are inaccurate, misleading, or even dangerous (14; 2; 27). For example, a bad actor could insert misinformation about a political candidate into a data source, which the RAG model could then present as factual information to a user. Another significant threat is indirect prompt injection, where an attacker embeds a malicious prompt within a document that is likely to be retrieved by the RAG model. When the model retrieves and processes this document, the hidden prompt can hijack the model's instructions, causing it to perform unintended actions, reveal sensitive information, or generate harmful content. This is a subtle but powerful attack vector that is unique to RAG architectures (5).

Furthermore, RAG models can amplify existing biases and toxicity present in their retrieval sources. If the external knowledge base contains biased language or harmful stereotypes, the RAG model will likely reproduce and even amplify these biases in its own responses. While standard LLMs are also susceptible to bias, the retrieval mechanism of RAG models can make this problem worse by actively seeking out and incorporating biased information from external sources (20).

### 2.3 Source Attribution

Source Attribution is the process of linking the generated text to the specific source documents that informed it. A straightforward approach would be through prompt engineering, where the retrieved documents are numbered within the prompt, and the LLM is asked to mention the document number for each sentence it generates. However, this method suffers from multiple limitations:

- **Forced Simplicity:** Forcing the model to cite a single source per sentence discourages it from performing complex reasoning that combines insights from several documents. It may oversimplify the answer to fit the rigid citation format.

- **Lost in the Middle:** For models with very large context windows, studies have shown they tend to recall information from the beginning and end of the prompt more effectively than information from the middle. A key source document placed in the middle of a long context might be overlooked for citation (12).

- **Brittleness and Lack of Reliability:** The LLM's ability to follow citation instructions is not guaranteed; it's a "soft" instruction rather than a hard-coded process.

## 3 Method

Due to the limitations of Source Attribution, we realized the need for a new method to understand how each document affects the LLM's response. To this end, we propose *Influence Score* (IS), a metric inspired by Partial Information Decomposition (PID) that aims to quantify the influence of each document on the generated response. We will first provide a brief introduction to PID, and then our proposed metric.

### 3.1 Partial Information Decomposition

PID (8) is a theoretical framework designed to analyze how a set of input variables provides information about an output variable. Its goal is to move beyond simple mutual information and dissect how the information is structured among the sources. The central idea is to break down the total information into distinct, non-overlapping components, each describing a different mode of interaction:

- **Mutual Information ($I$)** The mutual information between a source and the target variable.

- **Union Information ($U$):** The mutual information provided by at least one individual source.

- **Excluded Information ($E$):** The information in the union of the sources except one particular source.

Given source variables $X_1, X_2, ..., X_k$ and target variable $Y$, the PID is written as

$$E(X_i \rightarrow Y \,|\, X_1, X_2, ..., X_k) = U(X_1, X_2, ..., X_k; Y) - I(X_i; Y), \tag{1}$$

$$I(X_i; Y) = H(Y) - H(Y|X_i), \quad H(.) := \text{Shannon Entropy} \tag{2}$$

$$U(X_1, X_2, ..., X_k; Y) = \inf_Q I(Q; Y) \text{ such that } \forall i \, X_i \in Q, \tag{3}$$

where $E(X_i \rightarrow Y \,|\, X_1, X_2, ..., X_k)$ measures "What knowledge does the rest of the group have that $X_i$ is missing".

## 3.2 Influence Score

PID requires a complete probability distribution over all possible outcomes of the source and target variables to calculate mutual information. This requirement is impossible to meet in a RAG context, where neither the retrieved documents nor the generated response can be treated as traditional random variables. Specifically: 1. The retrieved documents are outputs of a retrieval function, not random events. 2. While the LLM's response is generated probabilistically, the sample space of all possible text outputs is too vast to define a complete probability distribution, making a true entropy calculation intractable.

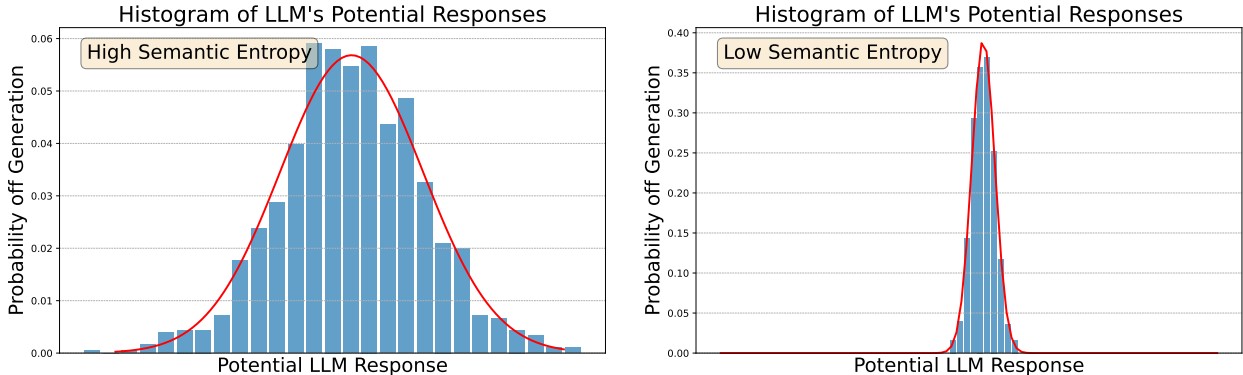

**Figure 1:** Visualization of an LLM's response probabilities in scenarios of high Semantic Entropy (left) and low Semantic Entropy (right). High entropy corresponds to low model confidence, while low entropy signifies high confidence.

As a workaround, we incorporate *Semantic Entropy* (3; 9; 15; 10), where the rationale is that sentences with similar meaning have a similar probability of generation. To be specific, Semantic Entropy quantifies the uncertainty in the meaning of an LLM's potential answers. It's calculated by generating multiple responses to a single query, clustering them by semantic similarity so that paraphrases are grouped together, and then computing the entropy over this distribution of meanings. A low entropy score signals model confidence, as the answers converge on a single meaning, while a high score indicates confusion, with responses spread across contradictory ideas. Figure 1 visualizes an LLM's probability distribution over possible response in the case of high and low Semantic Entropy.

In our context, the source variables ($X_1, X_2, ..., X_k$ are the retrieved documents (document $1, 2, ..., k$), and the target variable ($Y$) the generated response. The mutual information between the generated response and document $i$ is written as

$$I(X_i; Y) = H_S(Y) - H_S(Y|X_i), \tag{4}$$

where $H_S(Y|X_i)$ denotes the Semantic Entropy of the LLM response when augmenting document $i$ in the query, and $H_S(Y)$ the Semantic Entropy when no document is augmented.

For determining the Union Information, we need to find the total amount of information that can be obtained from at least one of the documents when generating the response. It is the information held by the collection of sources, and not the knowledge of the LLM itself. Therefore, we write the Union Information as

$$U(X_1, X_2, ..., X_k; Y) = H_S(Y) - H_S(Y|X_1, X_2, ..., X_k), \tag{5}$$

where $H_S(Y|X_1, X_2, ..., X_k)$ denotes the Semantic Entropy of the LLM response when augmenting all of the retrieved documents. Note a lower Semantic Entropy value for $H_S(Y|X_1, X_2, ..., X_k)$ relative to $H_S(Y)$ indicates that the LLM has grown more confident with the additional context, which means more information from the documents was taken into consideration.

We define the IS of document $i$ ($IS_i$) as

$$IS_i = -E(X_i \to Y \mid X_1, X_2, ..., X_k)$$
$$= I(X_i; Y) - U(X_1, X_2, ..., X_k; Y) = H_S(Y|X_1, X_2, ..., X_k) - H_S(Y|X_i). \tag{6}$$

Put into words, IS measures the change in Semantic Entropy of the LLM's potential outputs when the context is expanded from a single document $i$ to the full set of retrieved documents. This score reflects the degree to which the LLM prioritizes information from the other documents over the information in document $i$ alone. Overall, it quantifies the resulting shift in the distribution of meanings generated by the LLM.

The reasoning behind Equation 6 is that a lower Excluded Information value for document $i$ indicates greater reliance on that document by the LLM when generating a specific response. Therefore, a lower Excluded Information value should correspond to a higher IS. Moreover, a lower IS indicates a smaller Semantic Entropy value for $H_S(Y|X_1, X_2, ..., X_k)$ relative to $H_S(Y|X_i)$, which indicates a larger increase in the LLM's confidence when all documents are provided compared to when document $i$ is isolated. A larger difference implies that the material in the other documents is prioritized more than the material in document $i$.

## 4 Semantic Entropy

We derive semantic entropy by following these steps:

1. For each query, generate multiple ($N$) responses.

2. For each response, find its sentence embeddings. We use a Bert based model (4).

3. Next, calculate the similarity score between all the responses. We use the cosine similarity of the embeddings.

4. Normalize the scores to $[0, 1]$.

5. Next, estimate the probabilities by dividing each score by the sum of all the scores; that is, $p_i = \frac{\text{score}_i}{\sum_j \text{score}_j}$.

6. Finally, derive semantic entropy as $H_S = \sum_i p_i \log_2(p_i)$.

## 5 Experiments

We perform two sets of experiments. First, we perform malicious document identification, where we perform a poison attack and use IS to identify the poisoned document. Second, we conduct an ablation study and use a panel of judges (consisting of both human evaluators and an LLM) to validate that IS effectively identifies the most influential documents.

### 5.1 Malicious Document Identification

Our process uses the HotPotQA (25), Natural Questions (11), and MS MARCO (18) datasets for our queries and document database. First, we perform a poisoning attack on the retrieved documents to cause the LLM to generate incorrect responses. For each incorrect response, we then calculate our proposed metric, IS, for all of the retrieved documents. We consider our metric successful if the poisoned document ranks among the documents with the highest IS, since this demonstrates that IS can identify the most influential documents. Details of each step is provided below.

The rationale behind this experiment is that the poisoned document will introduce counterfactual data, resulting in lower model confidence and therefore a higher semantic entropy $H_S(Y|X_1, X_2, ..., X_k)$ in Equation 5, resulting in a lower $U(...)$. Similarly, the response generated in the presence of incorrect data is expected to differ from the unaugmented response more significantly than one generated with correct data, resulting in a higher $I(...)$ in Equation 4. Overall, a higher $I(...)$ and lower $U(...)$ will result in a higher IS, as seen in Equation 6.

**Database**: HotPotQA (25) is a question-answering dataset collected from the English Wikipedia. Its questions are constructed to require the introductory paragraphs of two Wikipedia articles to answer. Natural Questions (11) consists of real user questions submitted to Google Search and their corresponding answers extracted from Wikipedia. MS MARCO (18) is a question-answering dataset featuring 100,000 real Bing questions, each with a human-generated answer. For each question in these datasets, the five most relevant documents are identified and provided.

**Attack Method**: We implement a corpus poisoning attack similar to PoisonedRAG (28) to replace one of the retrieved documents provided by the datasets with a poisoned data to elicit an incorrect response. For each dataset, we submitted the queries until 1,000 incorrect responses were collected.

**Detection**: For each incorrect response, the IS for all of the documents are calculated, and documents are ranked from the highest to lowest. The summary of results are shown in Table 1 for each of the datasets and various LLMs. The variation in the rates can be attributed to the different degrees of overlap between the datasets and the LLM's knowledge. Although not within the context of this paper, it could be hypothesized

**Result**: Among all of the incorrect responses, in 86% of cases the poisoned document had the highest IS, in 95% of cases the poisoned document was among the top two documents, and in 100% of cases the poisoned document was among the top three documents. Let's consider the cases where the poisoned documents had the highest IS. A success rate of 86% among 3,000 samples translates to a *p*-value of less than 0.0001 and a confidence interval of 1.24%. The statistical significance of this result demonstrates that our metric is not acting randomly; rather, it is effectively identifying the poisoned document's influence on the incorrect response. Details on how to derive the *p*-value can be found in Appendix A. Compared to using prompt engineering approach to identify the poisoned document, our approach consistently achieves higher success rate. The exact prompt used is included in Appendix B.

### 5.2 Ablation Study

Our second experiment is a qualitative assessment of the Information Score (IS) performance:

- First, we generate a baseline response, **Response A**, by querying a RAG LLM with a full set of retrieved documents.

- Next, we calculate IS of each of the documents.

- We then create two new responses for comparison. **Response B** is generated using only the top two documents with the highest IS, while **Response C** is generated using the remaining $k-2$ documents.

- Finally, a judge compares both **Response B** and **Response C** to the original **Response A**.

- We consider the IS metric successful if the judge finds that **Response B** is more similar to **Response A** than **Response C** is, as this indicates that IS correctly identified the most influential documents.

**Table 1:** Success rate of our proposed metric, Influence Score (IS), in identifying the single poisoned document. The values show the percentage of cases where the poisoned document ranked in the top 1, 2, and 3 based on its IS after an incorrect response was generated. As a baseline, success rate of using prompt engineering approach to identify the poisoned document has been provided. The experiment is conducted on various LLMs and three datasets: HotPotQA (25) (HotPot), MS MARCO (18) (MS), and Natural Questions (11) (NQ).

| LLM | GPT-4 | | | LLAMA-3.3-70b | | | DeepSeek-R1 | | |
|---|---|---|---|---|---|---|---|---|---|
| Dataset | HotPot | MS | NQ | HotPot | MS | NQ | HotPot | MS | NQ |
| Top 1 | 84% | 77% | 87% | 92% | 87% | 86% | 89% | 85% | 86% |
| Top 2 | 92% | 88% | 85% | 100% | 96% | 95% | 98% | 94% | 95% |
| Top 3 | 100% | 100% | 100% | 100% | 100% | 100% | 100% | 100% | 100% |
| Prompt Eng. | 83% | 73% | 83% | 85% | 82% | 85% | 87% | 81% | 82% |

To elaborate, if **Response B** is more similar to the original **Response A** than **Response C** is, it proves the significant influence of the documents with the top two highest IS. This result shows that those two documents contribute more to the final output than all the other documents combined. Such an outcome would confirm that our proposed metric successfully measures the relative influence of each document on the LLM's response. We use **GPT-4** as the judge, and retrieve $k = 5$ documents.

Table 2 shows the rate that **Response B** was chosen over **Response C** on the HotPotQA (25), MS MARCO (18) and Natural Questions (11) datasets for various LLMs. The prompt given to the judge is provided in Appendix C. The results show that in almost all instances **Response B** was chosen, which indicates the success of our method in identifying the most influential documents.

**Table 2:** The rate that **Response B** (LLM response when augmenting the two documents with the highest IS) was chosen over **Response C** (response when augmenting the rest of the documents) to be more similar to **Response A** (response when augmenting all of the documents) by **GPT-4** as the judge. A higher rate indicates that IS correctly identifies the most influential documents. The experiment is conducted on various LLMs and three datasets: HotPotQA (25) (HotPot), MS MARCO (18) (MS), and Natural Questions (11) (NQ).

| LLM | GPT-4 | | | LLAMA-3.3-70b | | | DeepSeek-R1 | | |
|---|---|---|---|---|---|---|---|---|---|
| Dataset | HotPot | MS | NQ | HotPot | MS | NQ | HotPot | MS | NQ |
| Rate Response B is Chosen | 93% | 88% | 95% | 97% | 92% | 92% | 95% | 89% | 91% |

Next, we conduct the same experiment, but with human judges instead of an LLM. The experiments consists of 16 participants over 20 randomly chosen questions from HotPotQA. The participants were asked to decide whether **Response B** or **Response C** were more similar to **Response A**, and score their confidence on a scale of 1 to 5.

Figure 2 shows the the rate that **Response B** was chosen for each of the questions, and the "error bar" of each point represents the average confidence of the participants. To elaborate, a longer bar length represents lower confidence. We observe that in almost all cases the overwhelming majority chose **Response B**. Similar to the LLM judge, this indicates the success of our method.

Furthermore, looking at Figure 2 we notice that for some of the questions, the rate **Response B** was chosen is unusually low. However, upon further investigation and looking and the responses, we saw that the LLM

were able to respond to these question through internal knowledge and did not require augmented documents to answer the queries. As a result, **Response B** and **Response C** were both similar to **Response A**, and the participants had difficulty choosing one over the other.

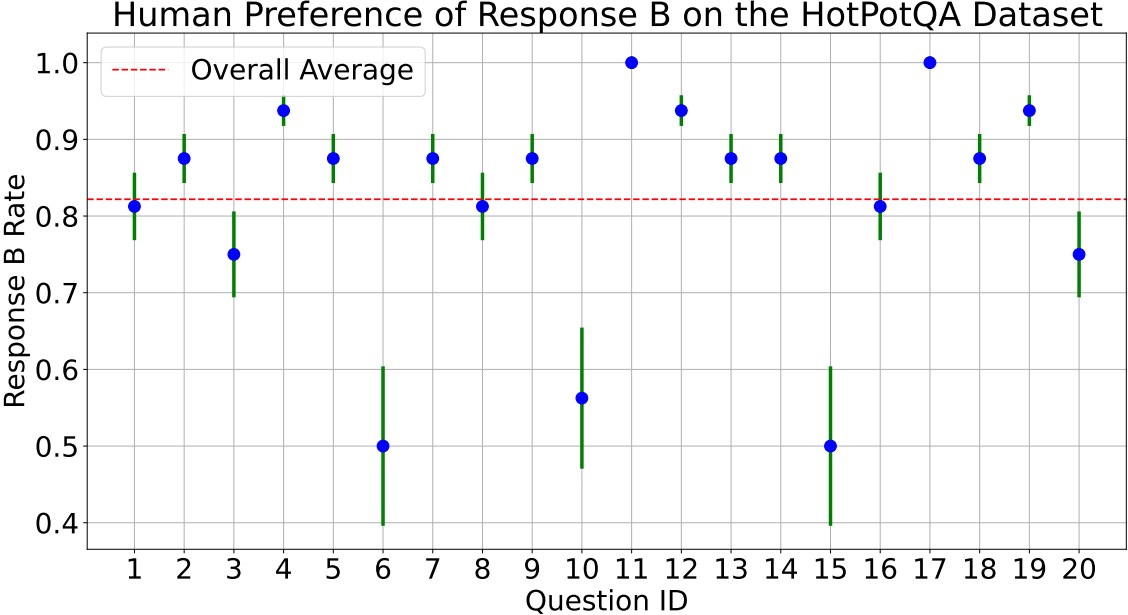

**Figure 2:** The rate that **Response B** (LLM response when augmenting the two documents with the highest IS) was chosen over **Response C** (response when augmenting the rest of the documents) to be more similar to **Response A** (response when augmenting all of the documents) by 16 participants as the judge. The questions are 20 randomly chosen from HotPotQA (25), and the error bars represents the average confidence of the participants' decision.

## 6 Drawbacks

Our approach introduces a drawback in format of additional computational cost. To be specific, to derive the IS for each of the $k$ documents according to Equation 6, the conditional semantic entropy needs to be calculated. To do so, the RAG LLM would need to be queried $2 \times k$ times: $k$ times using each document individually as context and another $k$ times using the set of all other documents as context. as explained in Section 4, the conditional semantic entropies are then calculated based on the similarity scores of these responses and the original LLM query, where all of the $k$ documents are augmented. This brings the total number of LLM queries to $2k + 1$.

## 7 Conclusion

In conclusion, this work addresses a critical explainability gap in Retrieval Augmented Generation systems. While RAG architectures enhance LLMs by grounding them in external data, they lack transparency, making it difficult to trace how retrieved documents shape the final output. This opacity can hide issues like factual inconsistencies, security vulnerabilities, and bias propagation, thereby limiting the reliability of RAG systems in high-stakes applications. To mitigate this, we introduced the Influence Score (IS), a novel metric designed to precisely quantify the contribution of each individual document to the generated response.

Our validation experiments robustly demonstrate the efficacy of the Influence Score. In a poison attack simulation, IS successfully identified the malicious document as the most influential source in 86% of cases, showcasing its potential as a tool for enhancing security and debugging. Furthermore, an ablation study confirmed that the documents ranked highest by IS are indeed the most critical for constructing a faithful

response. By isolating and measuring document-level impact, the Influence Score provides a vital mechanism for improving the transparency, security, and overall trustworthiness of RAG systems, paving the way for their more responsible and effective deployment.

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

## A Deriving $p$-Value

This section provides the details for deriving the $p$-value reported in Section 5.1

Our hypothesis ($H_a$) is that our introduced metric, Influence Score (IS), successfully identifies the poisoned document in cases when the LLM produces an incorrect response. To be specific, the are five retrieved documents, we calculate each document's IS, and rank the documents from high to low. In 86% of the cases, the poisoned document had the highest IS $\hat{p} = 0.86$.

The null hypothesis ($H_0$) is that our metric is random. The probability of selecting the poisoned document among five is 20% ($p_0 = 0.2$). We now find the $z$-score:

$$z = \frac{\hat{p} - p_0}{\sqrt{\frac{p_0(1-p_0)}{n}}} \sim 90.4, \tag{7}$$

where number of samples, $n$, is 3,000. a $z$-score of 90.4 corresponds to a $p$-value of less than 0.0001, indicating statistical significance.

## B Source Attribution Using Prompt Engineering

The prompt we used to identify poisoned documents using the prompt engineering approach for the experiment in Section 5.1 is provided below.

---

**Text Card B.1: Source Attribution Prompt**

**Instructions:** 1. Carefully review the user's query and the provided documents. 2. Synthesize an answer to the query using **only** the information found in the documents. Do not use any external knowledge. 3. After formulating the answer, determine which single document was the primary source of information for your response. 4. Provide your answer, and then, on a new line, cite the ID of the most relevant document in the specified format.

—

**[CONTEXT]**
**Document ID:** document id 1 **Content:** document content 1
**Document ID:** document id 2 **Content:** document content 2
**Document ID:** document id 3 **Content:** document content 3

—

**[QUERY]** user query

—

**[RESPONSE FORMAT]** Answer synthesized from the documents **Source:** Single most relevant Document ID

---

## C  LLM Judge Prompt

The prompt we used to as **GPT-4** to judge whether Response B or Response C is more similar to Response A used in Section 5.2 is provided below.

---

**Text Card C.1: LLM Judge Prompt**

Your task is to evaluate which of the two following responses, B or C, is more semantically similar to Response A.
[Response A]: RESPONSE A
[Response B]: RESPONSE B
[Response C]: RESPONSE C
Which response is more similar to Response A? **You must answer with only the exact text "Response B" or "Response C" and nothing else.** Do not provide any explanation, preamble, or punctuation.

---

