# OpenReview forum: "Quantifying Document Impact in RAG-LLMs"
_TMLR — Withdrawn by Authors_

### Review · Reviewer_Gpzc · 2025-11-13

**Summary Of Contributions:**

**Contributions:**

1. This paper introduces *Influence Score* (IS), a metric that estimates how much each retrieved document in a RAG system affects the generated response by measuring changes in semantic entropy under different document subsets.
2. It evaluates IS in two settings: a poison-attack simulation where the poisoned document is usually ranked most influential, and an ablation study where responses conditioned on top-IS documents are judged most similar to the full-context answer.

**Key Strengths:**

1. Addresses an important and underexplored problem: source attribution in simple RAG systems.
2. IS is model-agnostic and relies only on black-box access to the LLM, making it broadly applicable.
3. By design, IS is largely insensitive to document order and avoids the “lost in the middle” issue that affects prompt-based citation methods, since each document is also evaluated in isolation.
4. Overall writing and structure are clear.

**Key Weaknesses:**

1. The theoretical justification linking PID concepts to their semantic-entropy–based formulation is weak and not rigorously supported.
2. Many experimental choices (e.g., sampling hyperparameters) are under-specified or unjustified, limiting reproducibility and weakening the evidence.
3. Evaluation is narrow relative to the paper’s claims, with only a single, very simple baseline despite other alternatives.
4. The method is significantly more computationally expensive than simple approaches like leave-one-out.
5. Broader claims about transparency, safety, calibration, and bias mitigation are speculative and unsupported by downstream experiments.
6. Recent source-attribution algorithms are neither cited nor used as baselines.

**Additional Comments:**

I thank the authors for the effort they put into assembling this paper. The proposed approach has clear potential, and I appreciate the direction the work is aiming for. At the same time, I encourage the authors to conduct a more comprehensive empirical study to convincingly demonstrate that their technique is a strong alternative to existing source-attribution methods and is genuinely worth exploring further.

**Audience:**

Yes

**Audience Explanation:**

Yes, this is an interesting idea for source attribution in RAG that is worth exploring, and at least some TMLR readers would be interested in the findings, even if they’re preliminary. But the claims need to be properly validated first.

**Broader Impact Concerns:**

No concern!

**Claims And Evidence:**

No

**Claims Explanation:**

1. Authors claim that “we introduce the Influence Score (IS), a novel metric based on Partial Information Decomposition,” but the connection to PID is not theoretically sound. IS replaces Shannon entropy with a semantic-entropy proxy, which breaks the PID identities they rely on, and the paper provides no formal justification showing that IS inherits any expected information-theoretic properties. Also, key implementation details, such as the number of samples and sampling temperature, etc, are missing, making the method difficult to reproduce and weakening the theoretical grounding of the claim.

2. Authors introduce IS as an approach that helps with source attribution in general, but the evaluations are too limited to support this claim: they only compare against a trivial “cite the source” prompt and omit both simple baselines (e.g., leave-one-out) and stronger, recent source-attribution methods for RAG (e.g., Shapley-based document attribution as in Source Attribution in Retrieval-Augmented Generation, Nematov et al., 2025). In addition, the variability in their experiments is very limited: the second study mainly compares just two responses (top-2 IS docs vs. the remaining docs) with no exploration of top-1/top-3, random subsets, or correlations with answer correctness. Based on this, the claim that IS broadly “enhances source attribution” is not convincingly demonstrated beyond this narrow setup.

3. Authors claim that IS enables “Enhanced Source Attribution and Fact-Checking,” “Model Calibration and Identifying Bias,” “Document Relevance Ranking,” and “Adversarial Attacks and Model Poisoning”; however, the paper reports only two limited evaluations, without any downstream measurements of hallucination reduction, calibration (e.g., ECE/Brier), bias metrics, retrieval/ranking gains, or fact checking. As a result, these broader benefits remain unsubstantiated, and if not properly demonstrated, they should be presented as potential use cases rather than firm claims.

**Requested Changes:**

**Major changes:**

1. **Clarify or down-scope the PID claim + fully specify IS implementation details for reproducibility.**
   Either (a) provide a formal justification that the semantic-entropy–based IS still satisfies meaningful PID-style properties, or (b) clearly reframe IS as a heuristic influence score “inspired by PID” rather than “based on PID.” In both cases, fully specify the implementation details (number of samples, temperature/top-p, etc) so the method is reproducible.

2. **Strengthen the source-attribution evaluation and baselines.**
   Add both simple baselines (e.g., leave-one-out) and stronger recent methods (e.g., Shapley-based attribution for RAG, such as *Source Attribution in Retrieval-Augmented Generation* (Nematov et al., 2025)). Extend the second experiment beyond a single “top-2 vs. rest” comparison to include top-1/top-3 variants, random subsets, and correlations with answer correctness. To demonstrate that IS is a robust source-attribution method in general, additional experiments are needed; the authors can draw inspiration from *CONTEXTCITE: Attributing Model Generation to Context* and *Source Attribution in Retrieval-Augmented Generation* when designing these evaluations.


3. **Align broad claims with downstream evidence.**
   Either add downstream experiments that directly measure some of the claimed benefits (e.g., hallucination reduction, calibration metrics like ECE/Brier, bias metrics, retrieval/ranking quality, fact-checking performance), or explicitly reframe these as potential applications / future work rather than established outcomes.

---

**Minor Changes:**

1. **Clarify Figure 1 and fix axis labels.**
   The paper should explicitly describe how Figure 1 is generated, particularly how the x-axis (“Potential LLM Response”) is defined (e.g., response IDs, semantic clusters, or another indexing method). Additionally, the y-axis contains a typo and should be corrected from *“probability off generation”* to *“probability of generation.”*

2. **Correct the semantic entropy formula.**
   The entropy expression on page 5 is missing the negative sign. It should read:
   [
   H_S = -\sum_i p_i \log_2 p_i
   ]
   rather than
   [
   H_S = \sum_i p_i \log_2 p_i.
   ]

3. **Fix missing parenthesis and notation in Section 3.2.**
   The definition of source variables is missing a closing parenthesis. It should be written as:
   “the source variables (X₁, X₂, …, Xₖ) are the retrieved documents…”

4. **Correct subject–verb agreement and grammar issues.**
   Examples include:

   * “Details of each step **are** provided below.” not is
   * “The experiments **consist** of 16 participants…” not consists

5. **Improve phrasing in Section 6.**
   “Our approach introduces a drawback in format of additional computational cost.”
   should be rewritten as:
   “Our approach introduces a drawback in the form of additional computational cost.”

6. **Fix typos in Appendix A.**

   * “the are five retrieved documents…” → “**there** are five retrieved documents…”
   * Start new sentences properly (e.g., “A z-score of 90.4 corresponds…”).

7. **Clean up figure/table captions.**

   * Table 1: Add articles for readability (“As a baseline, **the** success rate of using **a** prompt-engineering approach…”).
   * Figure 2: Fix duplicated wording (“shows the the rate” → “shows the rate”).

---

### Review · Reviewer_6For · 2025-11-19

**Summary Of Contributions:**

**Summary Of Contributions:** The goal of this paper is to address the lack of a metric to quantify the contribution of retrieved documents to the outputs of RAG-based LLM generations. Specifically, it introduces a new metric called Influence Score (IS) based on information theory to assess the extent to which each retrieved document contributes to the final LLM response. The paper adopts Semantic Entropy, a popular framework for uncertainty quantification of LLM generations, to derive the Mutual Information between each retrieved document to the corresponding LLM response, then isolate the Union Information of all documents to define the final Influence Score. Empirical evaluations demonstrate that the Influence Score can detect malicious retrieved documents, as well as improving the quality of RAG-generated responses by identifying the most valuable reference materials.

**Strengths:**
- Attribution RAG-generated responses to retrieved documents is a relatively unexplored field, making it worthwhile to focus on.
- The proposed method is straightforward, and can be readily applicable to any RAG-LLM model.
- Experimental results demonstrate its effectiveness for detecting malicious document, and identifying high-valued documents for improving RAG generations.

**Weaknesses:**
- The proposed method incorporates Semantic Entropy as a major component for estimating the mutual information between retrieved documents and RAG-generated responses. However, the formulation of Semantic Entropy in this work is not properly motivated. Differently from [1], it is derived from the probability distribution over individual LLM responses rather than clusters of semantically equivalent responses. Similarity between sentence embeddings of LLM responses is not equivalent to semantic distance. For example, "The capital of France is Paris" and "The capital of France is not Paris" are clearly opposite in meaning, but have very high embedding similarity. NLI equivalence is a common tool for both uncertainty estimation [1] and LLM semantic evaluation [2], why is sentence embedding similarity preferable?
- Given the previous point, [1] proposes Discrete Semantic Entropy, which is directly applicable to your framework. Can additional experiment shows that the proposed Semantic Entropy is more effective for estimating Influence Score (IS)?
- Several implementation details of IS is missing: how to sample multiple responses from LLM? (e.g., temperature scaling, beam search, etc.); how many responses are necessary for effective IS attribution?
- There is no comparison with recent methods for RAG-generated response attribution. The paper propose an improved Source Attribution method, but lacks quantitative performance comparison with the original method [3], or recent Information Gain method such as [4].
- The paper lacks visual aids in general: a motivation figure comparing the proposed IS and original Source Attribution would significantly help readers understand the novelty of IS, some visual results on the ablation study are also useful to demonstrate the differences between Responses A/B/C and the extents IS attribute them to the retrieved documents

[1] Detecting hallucinations in large language models using semantic entropy (Farquhar et al. 2024)

[2] RAGAs: Automated Evaluation of Retrieval Augmented Generation (Es et al. 2024)

[3] Source Attribution in Retrieval-Augmented Generation (Nematov et al. 2025)

[4] InfoGain-RAG: Boosting Retrieval-Augmented Generation via Document Information Gain-based Reranking and Filtering (Wang et. al 2025)

**Additional Comments:**

N/A

**Audience:**

Yes

**Audience Explanation:**

RAG, attribution, and reliability of LLMs are active topics in the TMLR community, so researchers working on RAG systems, uncertainty, and interpretability would be genuinely interested in a method that quantifies how much each retrieved document influences the final answer.

**Claims And Evidence:**

No

**Claims Explanation:**

Overall, the experimental results show that Influence Score (IS) is effective and useful for detecting malicious retrieved documents and identifying high-valued documents to improve RAG generations. However, the current evidence is not fully convincing for the strongest claims, that IS is the preferred metric for document influence in RAG, mainly due to the lack of justification for its design, specifically the Semantic Entropy component using sentence embedding similarity, and the missing comparisons with recent methods for RAG-generated response attribution. See the **Weaknesses** above for more details.

**Requested Changes:**

1. Clarify and justify the Semantic Entropy formulation
- Provide a more thorough motivation for using sentence-embedding-based Semantic Entropy instead of the cluster-based definition in [1].
- Discuss explicitly why embedding similarity is an appropriate proxy for semantic equivalence, and discuss cases where it may fail (e.g., negation).
- Explain why NLI-based equivalence (as used in [1] and in works like [2]) is not used or is less suitable for your setting.
2. Empirical comparison of Semantic Entropy variants
- Compare Discrete Semantic Entropy from [1] against your proposed Semantic Entropy variant for IS calculation.
- Show how the choice of Semantic Entropy affects downstream metrics: e.g., accuracy of malicious document detection, attribution accuracy, and any RAG quality improvements.
3. Add missing implementation details and sensitivity analysis
- Clearly describe how multiple responses are sampled from the LLM: decoding strategy (temperature, top-p, etc.), number of samples, and any constraints on maximum length.
- Provide a brief sensitivity analysis: how does IS performance change as you vary the number of samples, temperature, or other key hyperparameters? This will help readers understand the practicality and robustness of IS.
4. Add baseline comparisons to recent attribution methods
- Quantitatively compare IS against: the original Source Attribution method [3], InfoGain-RAG [4]
5. Improve visual and qualitative explanations
- Add at least one high-level schematic / motivation figure contrasting IS with Source Attribution (e.g., how mutual information and union information are computed and interpreted).
- Provide visualizations from your ablation study showing how IS distributes influence across documents for different response types (A/B/C).

---

### Review · Reviewer_AgGj · 2025-11-19

**Summary Of Contributions:**

The authors introduce a new metric for measuring the contribution of a specific source document to the output of a Retrieval Augmented Generation (RAG) language model.

The metric, named the influence score (IS) of a document is the difference between the semantic entropy of a model request using the document only, and the semantic entropy of a model request using k different documents. The semantic entropy of a model request is computed by sampling N model outputs for this request, and calculating the entropy of the cosine similarity of their sentence embeddings.

The semantic entropy can be understood as the diversity of sampled generations, or a measure of the model confidence. A high IS therefore means that the model is more confident of its prediction when it retrieves from the document alone, than when it relies on the full corpus of k documents.

The authors provide two experiments for verifying the pertinence of the influence score:
1- when introducing a counterfactual source in the corpus, and forcing the model to generate an answer based on this erroneous data, they show that the IS clearly indicates the offending source as the main contributor to the (wrong) answer.
2- using k=5 sources for retrieval, they compare the model response (A) to requests using the 2 documents with the highest IS, (B) and the 3 others (C), and show that both an LLM and a panel of human judges agree that B is closer to A than C in a majority of cases.

**Strengths**
The proposed metric is interesting, and easy to compute
The experiments seem to confirm it works as intended

**Weaknesses**
There is no discussion of prior work, nor comparison to other approaches.
The method description is not clear: the introduction of many related quantities in section 3 hinders more than it helps, the last two paragraphs of section 3.2 are hard to parse.

**Additional Comments:**

I am not sure the term "poisoning attack" helps your argument. You are in fact trying to engineer a case where you can clearly identify a dependence between retrieval corpus and model prediction, by adding a counterfactual document in the retrieval set. A clearer explanation would probably help.

The link between what you propose and the second paragraph in the introduction is not clear to me. I don't think having a good attribution method (like IS) is central to addressing the problems presented in this paragraph (e.g. hallucination or bias propagation).

In figure 2, it seems that for three examples our of twenty, expert judgement is close to chance level (50%). Maybe the word "overwhelming" in the introduction is too strong.

In the poisoning experiment, how many documents were in the corpus (i.e. what is the value of k?), this would help assess the strength of your results.

**Audience:**

Yes

**Audience Explanation:**

The results are interesting and potentially useful. The method suggested can be applied in practice. The main issue is comparison with prior work (and some details about the methodology).

**Broader Impact Concerns:**

No impact concerns

**Claims And Evidence:**

No

**Claims Explanation:**

There is a large corpus of prior work on document attribution in RAG (see for instance the intro and related section of https://aclanthology.org/2024.emnlp-main.347.pdf), and the subject is still quite active (e.g. https://arxiv.org/pdf/2507.04480). None of these are discussed, or even mentioned, in the paper.  A related section and a discussion of prior work is really needed here, and the most significant measures of attribution should be used as baselines for the experiments. Without such comparisons, it is difficult to assess the claims made in the paper.

The methodology could be clarified. In particular, the description at the beginning of section 3.1 seems self-referent: "mutual information is defined as... mutual information... use formulas 5-7 in the definition). It is not clear to me that IS is an application of the PID framework. As you observe at the beginning of sec 3.2, the entropies in the PID should computed over an (unknown) distribution D of possible requests. Because these are hard to compute, you replace them by entropies of variables computed from output generated from a single request. I don't see how these two quantities are related. If they are not, wouldn't it be better to drop the reference to PID (or make it less central to your claim), and introduce the semantic entropy upfront? I believe this would reinforce the paper.

A number of additional information about the calculation must be provided. First, the calculation of the semantic entropy depends on a number of inferences (from a single request), which you denote as N. How large should N be? and how does the IS depend on the choice of N? (N may also have an impact on computational cost, which is not discussed in section 6). Second, the calculation of $IS_i$ depends on the retrieval corpus $X_1, ... X_k$ and in particular on its size, $k$. How are $k$ and the corpus chosen, how does this choice impact the calculation? A discussion of these choices; and of the dependences of IS on k, is really needed. (I see you use k=5 in the ablation, this seems very small).

**Requested Changes:**

* Discuss prior works and other methods for attribution. Add a proper related work section to the paper.
* Provide a comparison between IS and important alternative methods.
* Rewrite section 3, maybe simplify the methodology (e.g. since all results depend on semantic entropy, do we need the reference to PID? since the formula for IS only depends on $H(Y|X_i)$ and $H(Y|X_1 ... X_k)$, do we need the reference to mutual information which involved H(Y)?)
* Discuss the dependence of IS on k and the retrieval corpus. Some ablation experiments would be useful.
* Discuss the dependence on the estimation of semantic entropy on N, the number of inferences performed (maybe add it to the computational cost estimates of section 6).

---

### Note · Authors · 2025-11-26

I have read and agree with the venue's withdrawal policy on behalf of myself and my co-authors.